# Regulatory Small RNAs for a Sustained Eco-Agriculture

**DOI:** 10.3390/ijms24021041

**Published:** 2023-01-05

**Authors:** Selvaraj Barathi, Nadana Sabapathi, Kandasamy Nagarajan Aruljothi, Jin-Hyung Lee, Jae-Jin Shim, Jintae Lee

**Affiliations:** 1School of Chemical Engineering, Yeungnam University, Gyeongsan 38541, Republic of Korea; 2Guangdong Key Laboratory for Genome Stability and Disease Prevention, School of Medicine, Shenzhen University, Shenzhen 518060, China; 3Department of Genetic Engineering, SRM Institute of Science and Technology, Kattankulathur, Chennai 603 203, India

**Keywords:** bioremediation, biotic and abiotic, eco-agriculture, pesticide, small RNA

## Abstract

Small RNA (sRNA) has become an alternate biotechnology tool for sustaining eco-agriculture by enhancing plant solidity and managing environmental hazards over traditional methods. Plants synthesize a variety of sRNA to silence the crucial genes of pests or plant immune inhibitory proteins and counter adverse environmental conditions. These sRNAs can be cultivated using biotechnological methods to apply directly or through bacterial systems to counter the biotic stress. On the other hand, through synthesizing sRNAs, microbial networks indicate toxic elements in the environment, which can be used effectively in environmental monitoring and management. Moreover, microbes possess sRNAs that enhance the degradation of xenobiotics and maintain bio-geo-cycles locally. Selective bacterial and plant sRNA systems can work symbiotically to establish a sustained eco-agriculture system. An sRNA-mediated approach is becoming a greener tool to replace xenobiotic pesticides, fertilizers, and other chemical remediation elements. The review focused on the applications of sRNA in both sustained agriculture and bioremediation. It also discusses limitations and recommends various approaches toward future improvements for a sustained eco-agriculture system.

## 1. Introduction

Living organisms consist of various short RNAs, including microRNAs (miRNA), small interfering RNA (siRNA), piwi-interacting RNA (piRNA), small nucleolar RNAs (snoRNAs), transfer RNA (tRNA), and small regulatory RNA (srRNA). Non-coding RNAs, generally ~ 20–500 nucleotides in length, are grouped as small regulatory RNAs. These sRNAs are generated from long double-stranded RNA precursors or the RNA UTR region in the cell. They regulate the transcription and translation in prokaryotic and eukaryotic organisms [1] by inhibiting specific genes at the transcriptional or translational level [2]. sRNAs are associated with cell growth, differentiation, cell death, and cellular defense in eukaryotes [3] and the biotic and abiotic stress response in prokaryotes and eukaryotes.

A sustained eco-agriculture system is a new dimension that gives sustained farming a vision to protect the environment and biodiversity. The global demand for food and agricultural products has resulted in the adoption of various modern technologies, such as the usage of xenobiotics, which have proven to be hazardous to soil composition, life, and biodiversity [4]. Soil pollution by fertilizers, pesticides, and other xenobiotics of Polycyclic aromatic hydrocarbons (PAHs) used directly in agriculture and water pollution with various toxic elements (dyes and heavy metals) also form an indirect threat to soil fertility. In addition, the continued weakening of soil health due to new intrusions and pollution might lead to the deterioration of agricultural productivity and a loss of soil-mediated ecosystems [5]. Therefore, it is essential to meet the challenges, such as the growing need for food, degrading soil health, and loss of biodiversity and ecosystem, through effective and natural practices to achieve a sustained eco-agricultural system. Here, the applications of sRNAs in monitoring soil health, bacterial communities, and immunity against pathogens by the plants for sustained eco-agriculture are highlighted. The significant proliferation of pesticides led to groundwater and soil contamination, which remains an important hazard to agricultural ecology and human health [6]. Preservation of agricultural and horticultural products relies on chemical pesticides, but their improper usage has led to the depletion of the non-target organisms since most pesticides are made from highly toxic substances such as polycyclic aromatic hydrocarbons, atrazine, and paraquat [7,8]. In addition to direct environmental damage, pesticide toxicity is associated with various acute health conditions and even human death [9,10].

The microbial-based pesticides are an eco-friendly strategy to control the pest population [11]. Moreover, using microbes in the field promotes plant growth by facilitating various essential mineral solubilization [12]. Recently, the use of microbial-based products for pest control gained considerable attention but using whole microorganisms as a pesticide raised concerns of pathogenesis and distressing the non-target inhabitants; hence microbial sRNAs can be used as an alternative tool to overcome these limitations and extemporize the management of pests in an agricultural field. Sustained agriculture is farming in sustainable ways to meet the world’s demands without tampering the natural resources such as soil quality and the beneficial microbial community, whereas sustained eco-agriculture serves all the above-mentioned purposes with the addition of protecting the ecosystem. Bioremediation, restoration, and biomonitoring are the approaches that are used to achieve sustained eco-agriculture where small RNA can be used as tools. Regulatory sRNAs have proven roles in pest management, contaminant detection, biodiversity assessment, microbial community, environmental stress monitoring [13], promoting plant growth, protein degradation, and retort to biotic and abiotic stresses and signal transduction [14]. The manuscript extensively reviews the contribution of both microbial and plant small RNAs in monitoring soil health, bacterial communities, and immunity against pathogens for sustained eco-agriculture.

## 2. sRNAs in Plants

Small RNAs are essential for controlling plant gene expression at various stages of development, and under different conditions, sRNA-based virus defense is currently available in commercially permitted crops [15]. Plants synthesize numerous types of small regulatory RNA pathways to control gene expression patterns based on demand using highly conserved protein families (Figure 1). Small RNA-mediated gene silencing is induced in plants during pathogen invasion and unfavorable physical conditions to encounter abiotic and biotic challenges (Table 1). Dehydration, cold, and high salinity treatments significantly increased miR393 expression in Arabidopsis [16]. Overexpression of a rice miR319 gene (Osa- miR319) in transgenic creeping bentgrass (Agrostis stolonifera) increased tolerance to salinity and drought. siRNA-intermediated gene silencing only occurs when an infection has already invaded the host. Interestingly, some miRNAs appear to be involved directly in antiviral activity. For example, host plant sRNAs are engaged with antiviral or antibacterial immunity since they control resistance genes (R) or pathogenesis-related genes, and sRNA interceded specific cleavage is used to improve under normal conditions.

### 2.1. Abiotic and Biotic Stress Tolerance by sRNAs

In recent years, sRNA-mediated natural approaches have shown promising activity in plant defense against abiotic and biotic stresses. Drought and salinity are the main abiotic stresses that reduce crop productivity. Crops that withstand salt and drought have been developed using RNA interference (RNAi) technologies. Using the AtHPR1 promoter, RNAi-mediated down-regulation of farnesyltransferase in canola demonstrated increased resilience to seed abortion caused by a water shortage during flowering without compromising yield during drought [17]. To improve drought tolerance, the ubiquitin ligase gene OsDSG1 has been targeted by RNAi in rice [18]. Similarly, RNAi-mediated silencing of the C3HC4 RING finger E3 ligase OsDIS1 (for Oryza sativa drought-induced SINA protein 1) improved drought tolerance. The legume model plant Medicago truncatula was studied by Wang et al. [19], who exposed many drought-responsive miRNAs. These miRNAs’ expected and confirmed targets were engaged in a wide range of functions, including protein degradation, detoxification, plant cell growth, and transcription. Dehydration, cold, and high salinity treatments significantly increased miR393 expression [20].

Biotic stresses affect agricultural productivity and are extremely difficult to control owing to their rapid growth and environmental adaptations. miRNA-mediated tools have proven to be most effective against pathogenic bacterial strains. In Arabidopsis, miR393 represses the auxin production to resist Agrobacterium tumefaciens infections [21]. Fatty acids and their byproducts can hinder plant immunity. Thereby by eliminating the fatty acid desaturase (SACPD) gene by RNAi, Arabidopsis plants resist a variety of diseases [22]. The sRNAs (miRNA and siRNA) involved in the innate immunity of plants mediate the antiviral defense through gene silencing mechanisms. For example, recent reports showed that two miRNAs, nta-miR6019 and nta-miR6020, guided the cleavage of a TIR-NB-LRR immune receptor N to confer resistance to Tobacco mosaic virus (TMV) in the tobacco plant through the construction of trans-acting siRNAs (tasiRNAs) [23,24]. Moreover, the miR482*—*NBS-LRR regulation forms a classical feedback mechanism to avoid autoimmunity and conserve energy [25]. Plant sRNAs also support the pathogen-associated molecular patterns (PAMPs) to encounter immune-resistant pathogens.

**Table 1 ijms-24-01041-t001:** sRNA and their roles in plant defenses.

Name of sRNA	Host	Effector	Target	Role	References
Arabidopsis 22 nucleotide siRNA	*Arabidopsis*	Abiotic (nitrogen stress)	22 nucleotide siRNA	22 nucleotide siRNA inhibits the translation of specific genes and reduces the efficacy of protein change to deal with the stress of nitrogen deficiency.	[26]
miR12477	*Oryza coarctata*	Abiotic (Salinity)	LAO	LAO and oxidative stress regulates by Osa-miRNA12477 in plant salt tolerance	[27]
miR396	*Oryza*	Abiotic	GRF8	The target factors of referred genes resulted in inflamed grain size and raised brown planthopper resistance.	[28]
Muesultlberry 24 nucleotide SiRNA	*Botrytis cinerea*	Virus	MET1	24 nucleotide siRNA decreases the opposition gene methylation stages and expands the plant’s protection	[29]
miR165/166	*Arabidopsis*	Auxin	PHV and PHB	For the methylation of the PHV and PHB genes, complementarity between PHV and PHB mRNA and miR165/166 is anticipated.	[30]
miR393	*N. benthamiana* and *Arabidopsis*	Bacteria	MEMB12	The bacterial infection encourages the emission and accumulation of PR1 protein and pays to resistance.	[31]
miR477	Cotton plants	Fungus	CBP60a	CBP60a of mRNA divides by Ghr-miR477, facilitates the plant defense, and controls the biosynthesis of salicylic acid.	[32]
nta-miR6019	*Nicotiana tabacum*	Virus	Receptor N	Cleavage of transcripts of the Interleukin-1 and Toll receptor-NB-LRR protected receptor N from tobacco presents protection from tobacco mosaic infection.	[23]
miR2118	*Nicotiana benthamiana*	Bacteria and Virus	R gene	Mediated the novel layer of resistance against pathogen attack.	[33]
miR812w	*Oryza*	Fungus	LRR, ACO3, CIPK10	Overexpression of miR812w expanded protection from disease thru the rice impact *Magnaporthe oryzae*	[34]
TE-siR815	*Oryza*	Bacteria	WRKY45	Te-sir815 promotes the RdDM pathway’s transcriptional suppression of the key WRKY45 signaling pathway component, reducing rice tolerance to bacterial infection.	[35]
AtlsiRNA-1	*Arabidopsis*	Bacteria	AtRAP	A RAP domain protein implicated in disease resistance is changed by atlsiRNA-1.	[36]
miR398	*Arabidopsis*	Bacteria	COX5, CSD1, CSD2	miR398 negatively regulates disease resistance to bacteria and PAMP-induced callose deposition, which is also difficult for the miRNA directive in plant essential resistance.	[37]
dsRNA	*Nicotiana attenuata*	Hemiptera	HIGS	Relative to plant-processed sRNA, lengthy, unprocessed dsRNA has a higher efficiency.	[38]
hpRNA	*Maize*	Western corn rootworm	V-ATPase	V-ATPase subunit C showed less root damage through western corn rootworm	[39]
hpRNA	*Tobacco*	whitefly	V-ATPaseA	Improved whitefly resistance in transgenic tobacco by higher whitefly mortality and plant colonization	[40]

LAO: L-Ascorbate Oxidase; GRF8: Growth Regulating Factor; MET1: Methionine 1; PHV: Phavoluta; PHB: Phabulosa; MEMB12: Membrin 12; CBP60a: Calmodulin-Binding Protein 60a; LRR: Leucine-Rich Repeat; ACO3: 1-Aminocyclopropane-1-Carboxylic Acid Oxidase; CIPK10: CBL-Interacting Protein Kinase10; WRKY45: DNA-Binding Protein 45; AtRAP: Putative RNA-Binding Domain; COX5, CSD1, CSD2; HIGS- Host-Induced Gene Silencing; V-ATPase: Subunit C mRNA; V-ATPaseA: Subunit A mRNA.

### 2.2. sRNAs in Plant Defense Against Microbial Pathogens

Plants exhibit an alternate immune mechanism against microbial pathogens via identifying their pathogen-associated molecular patterns (PAMPs) that lead to the initiation of a downstream signaling cascade leading to PAMP-triggered immunity (PTI) [41]. A global sRNA profiling of microorganism interactions has given valuable data concerning the sRNAs involved with immunity and the adjusted expression of genes, and some sRNAs have even turned into the molecular marks of explicit PTI or ETI events. sRNAs generally support the plant immune system by either tuning the hormone networks or regulating the R proteins or plant immune regulator genes. The miR863-3p diminishes the transcripts of atypical receptor-like pseudokinase1 (ARLPK1) and ARLPK2, the negative immune regulators during the early stages of infection, and silences SERRATE, a positive regulator of the plant immunity during the late stage of infection to promote immunity against *P. syringe* [42] through sequential silencing and feedback inhibition. Plants can even disperse miRNAs to stop the fungoid pathogen *Verticillium dahlia* from expressing virulence genes [43]. RNAi-based crop defense methodologies have been utilized to control insects and pathogens [44], including bacteria such as *Agrobacterium*, fungi such as *powdery mildew*, and nematodes such as root-knot nematodes [45]. However, there have long been limitations to the genetically modified approach to crop protection, including limited societal approval in many markets and the impossibility of genetically changing most crop species [44]. As a result, much of the recent emphasis on RNAi for crop protection has switched toward non-transformative approaches [46].

### 2.3. sRNA Plant Defense in Pests

RNAi offers an opportunity to mimic or enhance that naturally derived pathogen control mechanism by delivering well-designed external dsRNA. Here, RNAi-mediated silencing strategies for pest control are carried out through Host-induced gene silencing (HIGS), spray-induced gene silencing (SIGS), and virus-induced gene silencing (VIGS). These methods can give long-term solutions for disease management, such as insects, viruses, and fungi. SIGS directly employs pathogen-gene-targeting dsRNAs or sRNAs, and VIGS uses the virus expression vector as the medium. Hyunkyu et al. proved that SIGS are effective at targeting diseases when sprayed directly on plants [47]. These innovative methods eliminate plant diseases and benefit from simplicity, high specificity, adaptability, and stability. It is important to highlight that the efficacy of SIGS for disease prevention is highly dependent on pathogen RNA uptake and secondary amplification of siRNA machinery, which is a limiting factor in this approach. Lately, a nanocarrier-based dsRNA distribution system has been developed to the specific targets and improve the sprayable quality of RNA pesticides [48]. The dsRNA technology preferred based strategy that could provide an eco-friendly method to control pests and reduce xenobiotics as an alternative to pesticide treatments (Figure 2). The use of this method is different, and it may vary depending on the environment. On the other hand, the regulation of dsRNA and its efficiency and constancy is not well recognized. Therefore, several practical features of the dsRNA application required more research, particularly for reducing the side effects on non-target microbes and animals.

When dsRNA is administered externally to plants, one hypothesized mechanism would be that plant cells can directly employ it to combat the pathogen via released vesicles holding the RNA at the point of infection and plasmodesmata [49]. Based on the target species’ sensitivity to RNAi, ability to activate the defensive system, and effectiveness of the delivery technique, the amount of RNA sprayed may change. Although RNAi is very comprehensive and specific, it can reduce crop pest outbreaks without harming beneficial insects or other field animals [50]. However, several factors, including the presence of the cell wall, plant-specific procedures, injury to biomolecules during application due to external environmental conditions, and tissue damage to plants resulting in necrosis and browning, limit the deployment of spraying techniques. The advent of nanocarriers as a vehicle for sRNAs in plants has yet to be studied. It opens future avenues of research to reduce the prevalence of pests and pathogens in significant cultures around the globe. Plant-derived extracellular vesicles (EV), or exosomes, are potentially considered vehicles for carrying the sRNA species [51]. However, our understanding of plant EV is inadequate, and the ability of artificial EVs to perform in vivo still needs to be discovered. Since dsRNA used in treatments decays quickly (30 or 72 h) in water and soil, they may not pose an environmental danger [52,53]. It holds extraordinary potential in pest control. These discoveries propose that sRNA conjunction with based pest control approaches, could be competent in overcoming these toxic pesticides to establish sustained agriculture.

## 3. sRNA-Based Xenobiotic Biosensors

The rapid growth of the population and industries have led to agricultural pollutants that damage the environment [54]. Therefore, monitoring toxic compounds and heavy metals in the atmosphere is crucial to maintain sustained agriculture. Table 2 lists the advantages and disadvantages of various methods used frequently to recognize environmental pollutants. Enzyme sensors have been used to estimate toxic compounds by enzymatic reactions; for example, peroxidases, laccase, and tyrosinase are used as biosensors for detecting and degrading phenolic compounds [55]. Enzyme sensors have also been used to evaluate heavy metals as they can bind to particular proteins. An advanced model of biosensors for the recognition of heavy metals and pesticides was developed, exploiting their inhibitory properties on horseradish peroxidase and glucose oxidase [56]. Table 3 lists several biosensors identifying pesticides and heavy metals in environmental samples.

## 4. Molecular Approaches to Combat Xenobiotics of Agri-Ecosystem

Industrialization and development have led to the massive accumulation of xenobiotics in natural environments, leading to soil and water pollution [81]. Synthetic pesticides are a prime example of xenobiotics, which have consistently been used in agriculture and insect control. There is a need to control pesticide use and naturally induce plant immunity against pests, pathogens, and parasites. This section outlines the recent advances and comprehensive insights about microbial populations, their mechanisms of interaction with contaminants, their metabolic activities, and aspects of molecular biology and genetic regulation that are attained through ‘omics’ approaches [82]. Genomics and metagenomics are powerful tools that allow environmental microbiologists to have a wider range of options for comprehending inhospitable microbial ecosystems [83]. For instance, the relative abundance of the mRNA and the corresponding sRNA are vital for evaluating gene silencing networks. Therefore, to extract the superior gene of community mRNA (meta transcriptome) from different environmental samples, various genomic and metagenomic molecular methods are employed where these techniques are suitable for capturing new mRNA transcriptions from ecological microbial networks. Combining these state-of-the-art molecular technologies with bioinformatic approaches makes it possible to better understand unknown soil microbial communities and the mechanisms underlying potential bioremediation.

### 4.1. Monitoring the Microbial Communities and Bioremediation Processes In Situ

In situ molecular approaches reveal the cellular rRNA composition, indigenous microbiota’s structural dynamics, and native physiological status under various environmental settings. Direct in situ probing with gene sequences or rRNA targeting in a combination of autoradiography and cell counting approaches can be employed to evaluate the impact of environmental factors or else agitation on the native ecophysiology of the microbial community [84]. Likewise, molecular microbial bioreporters or biosensors for in situ environmental pollution detection have recently been developed.

### 4.2. RNA-Based Analysis

The RNA analysis through meta-transcriptome sequencing is most valuable in evaluating the connections between the ecological conditions in a microbial territory and specific in situ activities of native microorganisms [85]. For example, the meta-transcriptome study done in the samples of large mouse intestine, cow rumen, kimchi culture, deep-sea thermal vent, and permafrost revealed their taxonomic relationships and their uniformity towards amino acid, nucleotides, and glycan degradation pathways [86]. Moreover, in situ analysis of RNA molecules led to the discovery of the degenerate initial gene sets associated with two groups of dioxygenase genes, *dntAc* and *ndoB*, that were earlier replicated from *Burkholderia* sp. strain DNT and *Pseudomonas putida* NCIB 9816-4, respectively. The difference display is another powerful RNA-based tool used extensively to examine eukaryotic gene expression, which is modified to measure bacterial rRNA diversity [87]. These expression studies were employed to investigate the specific gene regulation in the anthropogenic environment conditions [88] and to compare the specific expression between the gene families under diverse environmental conditions. Thus, the RNA-based meta-transcriptome and differential display analysis can reveal taxonomic, community, and functional signatures of the microbiome, which can be used against environmental hazards.

### 4.3. Method of Microbial Community Fingerprinting

PCR-based genotypic fingerprinting strategies for checking microbial networks [84] to profile microbial population dynamics during ongoing bioremediation processes have recently allowed researchers to find “key microbial strains” that are critical for the success of ex/in situ bioremediation. Molecular methods have emerged as an important tool for managing microbial communities and establishing optimal operating conditions, thereby increasing the potential of bioremediation systems [89]. There are several online automatic fragment length taxonomic assignment tools, such as torast (http://www.torast.de), TAP-TRFLP (http://rdp.cme.msu.edu), and MiCA (http://mica.ibest.uidaho.edu), have been established to achieve in silico T-RFLP techniques of 16s rRNA gene sequences available in the databases. Sequence analysis of the 16S rRNA gene can offer an in-depth investigation of the microbial population by specifically amplifying and sequencing the hypervariable sections of the gene [90]. This is used to identify new, uncultivable, or phenotypically unknown microorganisms (Clarridge, 2004). The phylogenetic order of the bacteria involved in the bioremediation process can be determined using 16S rRNA sequences taken from polluted sites. Single-strand conformation polymorphism (SSCP) is a genotyping technique that employs the three-dimensional structure of the single-stranded rRNA gene, which may be directly related to differences in the rRNA gene sequence. This method examines the electrophoretic mobility of single-stranded rRNA genes under non-denaturing conditions, and the resulting band patterns are used to differentiate between microbial phylogroups. Similarly, an additional online program, TRiFle, can simulate and produce T-RF datasets with arbitrary sets of DNA sequences from specific targets (e.g., genes involved in any metabolic pathways) or unpublished sequences [91,92]. Additionally, there are numerous online programs, including phylogenetic assignment tool (PAT), TRUFFLER, and APLAUS, that can be used to compare T-RFs predicted from an analysis of rRNA database sequences with the structure of the microbial community [93,94] used T-RFLP to recognize the temporal microbial community dynamics during the bioremediation of oil-contaminated Antarctic soil. The major advantage of these T-RFLP techniques is the demonstration of multiplex T-RFLP (M-TRFLP), which is valuable for the immediate profiling of many taxonomic groups of micro-organisms (two to four different taxa) in an ecosystem. This study aims to reveal that molecular biology techniques have been widely applied in environmental research as a basis and source of help for monitoring devices, pollution management, and environmental health.

### 4.4. Fluorescence In Situ Hybridization (FISH)

Fluorescence in situ hybridization has been used to recognize microbial architecture associated with onsite bioremediation processes [95]. FISH methods are susceptible, even at the single-cell level. FISH is a taxonomic method frequently used to assess whether members of a specific phylogenetic association are present; it allows for direct viewing of uncultured microorganisms and quantifying exact microbial structures. The use of FISH solely does not reveal any information about the metabolic activity of bacteria. However, FISH techniques can be combined easily with other methods (nanoSIMS, mRNA FISH, and microautoradiography) to analyze the functionality of certain microbial communities and for biological sensing of pollution in a critical environment [96]. In a FISH experiment, several group-specific rRNA probes targeting prokaryotic and eukaryotic microbial taxa can be employed for simultaneous phylogenetic categorization and quantification of physiologically active microbial communities in an environmental sample. As a result, the combination of FISH with microautoradiography (FISH-MAR) allows for both phylogenetic and functional identification of substrate-active cells within complex microbial communities. The FISH-MAR technique involves incubating an environmental sample for a short p with a radioactively labeled substrate, followed by FISH identification of microbial populations and in-parallel processing of identified microbial cells with radiation-sensitive photographic silver emulsions. Due to the simple accessibility of sludge biomass for fixation, staining, and hybridization tests, the FISH-MAR approach has been most frequently utilized to discover important biodegradative microbial phylotypes within activated sludge systems. The objective of this section is to demonstrate the widespread use of molecular biology techniques in environmental research as a foundation and source of assistance for monitoring devices, control of pollution, and bioremediation.

## 5. sRNAs as a Tool for Future Green Environment

Plants have a diverse range of sRNAs, which rigorously manage gene expression and protect plants from hazardous exterior environments (Wang and Chekanova, 2016). These systems of plant sRNAs provide incredible value to plants to initiate and restrict the expression of essential stress regulatory genes by epigenetically directing mRNA constancy and interpretation during the comprehensive rewiring of gene expression expected to endure stresses [97]. Recent studies in microbes, plants, and animals find new mechanisms of sRNA activity: by PTGS and epigenetic regulation, facilitating double-strand breaks (DSB) repairs and through the elements of such-siRNAs that mask the intronic *cis*-elements under stresses in plants [98]. Moreover, it has been shown that an sRNA-mediated response works either by swiftly enhancing the translation of protective factors compared to untreated plants or by killing the target pests. Plants release interspecies sRNAs to expand the defense system to neighboring plants and silence pathogenic microbial mRNA. Plants use extracellular vesicles to transport their secreted sRNAs to prevent abiotic environments from degrading them [99]. Evidence suggests that sRNA-based approaches are latent to control insects, microbial pathogens, and pesticide agents. In addition, they are easily degraded and can be a replacement for chemical pesticides. In addition, agricultural fields can gain biodiversity and increase soil fertility. Overall, the application of sRNA is a promising solution for pest problems as its eco-friendly and specific.

Microbial sRNAs (miRNAs) are most significant because they are vital players in controlling gene expression in bacteria. sRNA-mediated gene regulation is more viable than regulation by protein-based mechanisms or transcription factors [100]. Importantly, under toxic environmental conditions, sRNAs effectively participate in quickly reprogramming the cell metabolism by controlling the expression and constancy of a few objective mRNAs [101,102]. The capability of sRNAs to sense environmental changes rapidly is a significant feature compared to regulating stress genes when the microenvironment is variable and unfriendly.

The distinguishing outline dedicated to genome editing, as a feature of the bacterial versatile immune response system, CRISPR, has turned into the Swiss armed force blade of genetics, with assurances and difficulties in biotechnology. Similarly, sRNA-mediated gene regulation has a wide range of applications that needs extensive exploration. Nevertheless, some technical problems should be settled. For instance, substrates for industrial fermentation can be polluted with diverse microorganisms. Such impurities can prevent the development of ideal (dsRNA-expressing) microbes and decrease the effectiveness of the fermentation process, hence decreasing efficiency [103]. These issues in this field, dynamically designed microorganisms for sRNA production, and inexpensive purified sRNA will open up greener agribusiness without requiring synthetic pesticides to defend plants from insects and microbial diseases. In addition, significant developments in the field of sRNAs have resulted from the routine application of high-throughput RNA-seq. To fully realize this method’s promise for development, the experimental and analytical issues, especially the inflexible limit of the RNA sequencing space, must be solved. Based on it, we can learn more about their remarkable diversity, evolution, and, most importantly, their roles in the field of sustainable agriculture.

## 6. Conclusions and Future Perspectives

Current studies recognized that sRNAs are major molecular regulators of plant and microbial defense responses that work under stressful conditions. Specifically, sRNAs are developing as the next-generation tools for enhancing resistance to biotic and abiotic conditions and detecting and monitoring the toxic substances in agri-ecosystem. Moreover, these discoveries provide new ways for crop protection, disease management, and bioremediation processes. The efficiency of sRNA uptake by nematodes, fungal and bacterial pathogens, and pests proposes that this expertise could be applied to control animal pathogens to improve livestock and human health if mammalian fungal pathogens can similarly take up external RNAs. The sRNA sensing capabilities are high and accurate, and the future trend will be focused on these biochemical features for developing biosensors.

## Figures and Tables

**Figure 1 ijms-24-01041-f001:**
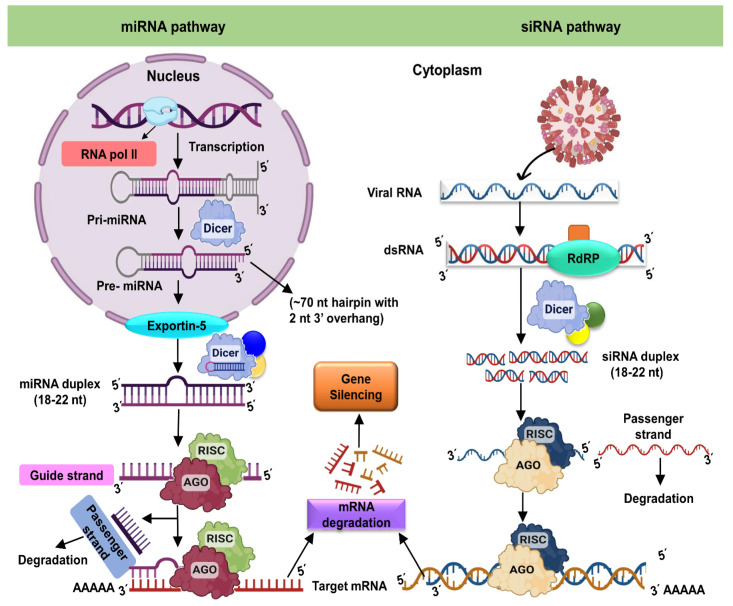
Mechanism of RNA interference in plant cells. The miRNA pathway, on the left, starts in the nucleus via transcription, resulting in the helix folding primary miRNA (pri-miRNA). Dicer first cleaves the pri-miRNA in the nucleus to produce the precursor miRNA (pre-miRNA), which Dicer then cleaves to produce the roughly 22-nucleotide miRNA duplex. The duplex is transported into the cytoplasm and loaded into the RISC. In RISC, one strand of the miRNA duplex (the passenger strand) is eliminated while another strand (the guide strand) is maintained. When the RISC-bound guide RNA attaches to a target RNA, it causes cleavage or translational arrest. When a target RNA (typically a structural aberrant RNA or those from an infected plant virus, as in this case) is detected by Dicer, the target RNA is cleaved into short (21–24 nucleotide) duplex RNAs. In the case of duplex miRNAs, one strand (the passenger) is delivered and degraded while the other (the guide RNA) is maintained by RISC. Cleavage occurs whenever the RISC-bound guide RNA attaches to a target RNA. The RNAi response within the cell can be amplified by RNA-dependent RNA polymerase activity. AGO: Argonaute protein; DCL: Dicer-like protein; RISC: RNA-induced silencing complex; RdRP: RNA-dependent RNA polymerase; RNAi: RNA interference.

**Figure 2 ijms-24-01041-f002:**
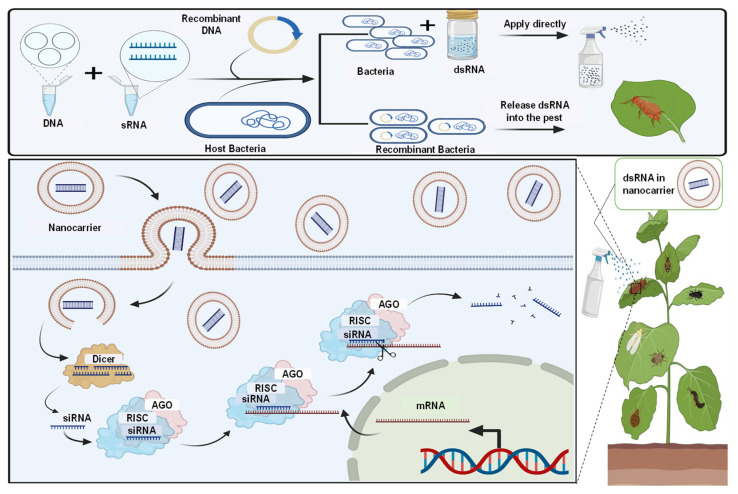
Microbial dsRNA production system and their technology to advance plant production. The dsRNA produced by microbes can be used directly for pest management by the nanocarrier-based transdermal dsRNA transfer system, and spraying can aid in developing sprayable RNA pesticides.

**Table 2 ijms-24-01041-t002:** Methods Used for Detection of Environmental Pollutants.

Detection Methods	Advantages	Disadvantages
Physical	Erosion, aeration, runoff, infiltration rate, and water holding capacity are always associated with certain hydrological processes [57]—no ethical clearance is needed.	Not giving standard results when applied in higher surface runoff and small water infiltration. Physical properties—a toxic agent Slow and cost-effective [58]
Chemical	Easy assessment with quick results. Both are financially profitable and efficiently capable of running in high-level pollutant conditions [59]. There are no ethical issues.	Involves an oxidation stage assuming the metals are complex High mud construction, dealing with and removal issues (the board, treatment, price). Low restrictions of detection and produced secondary hazardous intermediates [60]. Samples need cleaning before starting the process usually.
Electrochemical detection	Anthropogenic contaminants such as pesticides and Heavy metal ions contribute significantly to versatility. They are attributed to their greater sensitivity and inequitable ability. Fast analytical response and predictability in the process [61].	Low cost with acceptable reproducibility remains to be challenging.
Nano-biomaterials	One of the leading biosensors using green synthesis and nanofabrication technology. Precise and efficient detection and also a small size [62]. Helps to maintain the environmental sustainability Decrease waste production and eco-friendly techniques.	Possible to reduce the stability under highly toxic chemical conditions. Low efficiency in severe contamination conditions [63].
Bacteria	Inexpensive. Accessible to high-throughput formats and flexible to moveable devices. Cost-effective and straightforward to handle. Results are possible within hours.	Probable ethical questions about consuming genetic modifications [64]. Needs distinct apparatus for sterilized work. Maintenance is hard.
Algae	Not affected by toxic substances in the immobilized form. Robust and more reproducible. Simple and budget-friendly. Very quick (hours or days). Accessible to high-throughput formats [65]. It regulates the total toxicity of the sample.	Algae senses can detect only a particular set of toxic substances. Supplements in complex samples might mask the impacts of toxins.
Yeast	Mainly genes are possible. Quick results may be possible (Hours or days). Transfection with fully functioning vertebrate. User-friendly devices [64].	Maintaining sterile equipment for work. Unicellular organism.
Enzyme	A few of the enzymes, such as tyrosinase, peroxidases, and laccase, assist the growth of biosensors for degradation of a specific compound, such as phenolics, and utilizing different microorganisms also in free-state or immobilized structures.	Long time duration for recovery and little significance for the whole organism [66].
Tissue explants	Opportunity to use excess tissues as butchers.	Weakening of tissues after a comparatively short time. Do not reproduce general factors.
Animals in vivo	The most exact detecting structure for conversion of results to human.	Cost-effective and ethical issues.

**Table 3 ijms-24-01041-t003:** Important biosensors developed for the detection of certain toxic compounds.

Pollutant Analysis	Biosensing Elements	Detectors	Sources
Pesticides			
Paraoxon	Phosphotriesterase [67]	Optical	Medical samples
Methyl parathion	Organophosphorus hydrolase [68]	Electrochemical	Wastewater and soil
Atrazine	Tyrosinase [69]	Amperometric	Wastewater and soil
Dichlorvos	Choline oxidase [70]	Amperometric	Soil
Ametryn and acephate	E. coli, Bacillus subtilis, and S. cerevisiae [71]	Electrochemical	Soil
Fenobucarb	Glutathione S- transferase [72]	Bioluminescence	Soil
Heavy Metals			
Cadmium, lead, and copper	Glucose oxidase [73]	Electrochemical	Soil
Urea, organophosphates, and ethanol	Shewanella oneidensis [74]	Electrochemical	Medical samples
Nickel, copper, cadmium, and zinc	Horseradish peroxidase [73]	Luminometer	Tap water
Chromium	Glucose oxidase [75]	Optical	Water
(VI) and (III)	*S. cerevisiae* [76]	Electrochemical	Wastewater
Nickel, cadmium, copper, and zinc Cr(III), nickel, copper, cadmium and zinc	*E. coli* [77]	Electrochemical	Activated sludge
Mercury (Hg+2)	DNA [78]	Electrochemical	Soil

sRNA-based biosensors are rarely investigated in detecting environmental pollution, but it is evolving continually as an efficient tool in maintaining sustainable agriculture. Of interest, a few sRNAs have significant sensitivity to identify and respond to molecular and toxic substance signals and regulate active modulation of gene expression via RNA interference, with diverse mechanisms affecting stability, and splicing, including translation and transcription. In addition, toxicity assessments on the interaction of toxic compounds with DNA molecules have been established, especially for toxicity (PAHs) screening assays and monitoring environments. A recently established model of DNA-based biosensors can detect ten different derivatives of 1,3,5-triazine herbicides [79]. However, research on further developing small RNA detection specificity has yet to be made available. Besides using small RNA molecules from the original sources, the idea features a potential to construct synthetic cell systems focused on applications in environmental remediation, diagnostics, and next-generation therapeutics. Indeed, the report by [80] describes the development of an artificial fluorescent RNA-based aptamer biosensor for detecting the chemical compound theophylline, which is converted to a fluorescent signal. In this report, we propose a new perspective to combine the ‘omics’ data with ‘synthetic biology’, to synthesize RNA or DNA-based aptamers that mimic the sRNAs of plants and microbes that recognize sensing biohazard elements such as PAHs compounds and heavy metals in the environment. This approach would develop effective, controlled RNA/DNA biosensors with above 95% specificity and reliability.

## Data Availability

Not applicable.

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
