# Peer review of "Regulatory Small RNAs for a Sustained Eco-Agriculture"

_ijms, 2023, doi:10.3390/ijms24021041_

Round 1

Reviewer 1 Report

1.     Line 83: what is “R or pathogenesis-related genes”? A typo?

2.     The Latin name of plants, bacteria and viruses need to be italics.

3.     I think it is worthy to have a section describe sRNAs in plant and the communication of sRNAs between plant host and pathogen in detail. This would lay down foundation for the application of sRNA in agriculture.

Author Response

Detailed responses to the Reviewers’ comments

Thank you very much for your kind comments. We will try our best to revise the manuscript in order to improve its quality according to your suggestions and suit the requirements of the journal. Below are the itemized responses from authors to the comments of editor/reviewers (BLUE – Comments; Black – Authors response; Red – Revised text).

Reviewer #1:

  1. Line 83: what is “R or pathogenesis-related genes”? A typo?

             Resistance genes (R-Genes) are genes found in plant genomes that produce R proteins that confer pathogen resistance to plants (Line 91).

  1. The Latin name of plants, bacteria and viruses need to be italics.

            Thanks for your valuable comments. All the above-mentioned comments are revised

  1. I think it is worthy to have a section describe sRNAs in plant and the communication of sRNAs between plant host and pathogen in detail. This would lay down foundation for the application of sRNA in agriculture.

            According to the reviewer suggestions, sRNA in plant and their roles are explained by section and revised text are highlighted in red color (Line 159-175).

2.2. sRNAs in Plant Defense Against Microbial Pathogens

Plants exhibit an alternate immune mechanism against microbial pathogens via identifying their pathogen-associated molecular patterns (PAMPs) that lead to the initiation of a downstream signaling cascade leading to PAMP-triggered immunity (PTI) [41]. A global sRNA profiling of microorganism interactions has given valuable data concerning the sRNAs involved with immunity and the adjusted expression of genes, and some sRNAs have even turned into the molecular marks of explicit PTI or ETI events. sRNAs generally support the plant immune system by either tuning the hormone networks or regulating the R proteins or plant immune regulator genes. The bacterial pathogen Pseudomonas syringae, which carries a variety of effectors, induces miR863-3p. The miR863-3p diminishes the transcripts of atypical receptor-like pseudokinase1 (ARLPK1) and ARLPK2, the negative immune regulators during the early stages of infection, and silences SERRATE, a positive regulator of the plant immunity during the late stage of infection to promote immunity against P. syringe [42] through sequential silencing and feedback inhibition. Plants can even disperse miRNAs to stop the fungoid pathogen Verticillium dahlia from expressing virulence genes [43]. RNAi-based crop defense methodologies have been utilized to control insects and pathogens [44], including bacteria like Agrobacterium, fungi like powdery mildew, and nematodes like root knot nematodes

Reviewer 2 Report

In this study, the authors considered a number of different aspects of the use of small RNAs for plant protection, as well as for the monitoring of various xenobiotics. The use of miRNAs and their role in protection are fairly well described. However, the review almost does not discuss the delivery of small RNAs to their target. How should this happen? I believe it would be desirable to include another subsection in the review, discussing how small RNAs can be delivered to their targets. This is especially true for pests and pathogens. Figure 2 is not enough. In general, the review is well written and can be accepted for publication in IJMS after minor revisions.

Some small remarks.

References must be made in accordance with MDPI rules.

Lines 28-29. Too many abbreviations. Must be decrypted.

Lines 102-103. pri-miRNA or pre-miRNA?

Line 162 Reference must be at the end of the sentence, numbered in square brackets

Line 173 The same.

Author Response

Detailed responses to the Reviewers’ comments

Reviewer #2:

Thank you very much for your kind comments. We will try our best to revise the manuscript in order to improve its quality according to your suggestions and suit the requirements of the journal. Below are the itemized responses from authors to the comments of editor/reviewers (BLUE – Comments; Black – Authors response; Red – Revised text).

In this study, the authors considered a number of different aspects of the use of small RNAs for plant protection, as well as for the monitoring of various xenobiotics. The use of miRNAs and their role in protection are fairly well described. However, the review almost does not discuss the delivery of small RNAs to their target. How should this happen? I believe it would be desirable to include another subsection in the review, discussing how small RNAs can be delivered to their targets. This is especially true for pests and pathogens. Figure 2 is not enough. In general, the review is well written and can be accepted for publication in IJMS after minor revisions.

According to the reviewer suggestions, we discussed the sRNA delivery systems to their targets.

2.3. sRNA Plant Defense in Pests

                When dsRNA is administered externally to plants, one hypothesized mechanism would be that plant cells can directly employ it to combat the pathogen via released vesicles holding the RNA at the point of infection and plasmodesmata [49]. Based on the target species' sensitivity to RNAi, ability to activate the defensive system, and effectiveness of the delivery technique, the amount of RNA sprayed may change. Although RNAi is very comprehensive and specific, it can reduce crop pest outbreaks without harming beneficial insects or other field animals (Line 206-212).

We are very much thankful to the reviewer for the positive response on our work and providing

valuable suggestions. As suggested, we have addressed all the issues. The point-wise response to all the comments are appended below:

  1. References must be made in accordance with MDPI rules.

All the references are checked once again and corrected based on the MDPI format

  1. Lines 28-29. Too many abbreviations. Must be decrypted.

MicroRNAs (miRNA), small interfering RNA (siRNA), piwi-interacting RNA (piRNA), small nucleolar RNAs (snoRNAs), transfer RNA (tRNA) and small regulatory RNA (srRNA) (Line 29-31).

  1. Lines 102-103. pri-miRNA or pre-miRNA?

The miRNA pathway started in the nucleus via transcription, resulting in the helix folding as primary miRNA (pri-miRNA). Dicer first cleaves the pri-miRNA in the nucleus to produce the precursor miRNA (pre-miRNA) (Line 126-128).

  1. Line 162 Reference must be at the end of the sentence, numbered in square brackets

Line number 162 references are revised as per the reviewer suggestions (Line 212).

  1. Line 173 The same.

Line number 173 references are revised as per the reviewer suggestions (Line 222).

Reviewer 3 Report

The manuscript aims to provide a review concerning the applications of small RNA for "a sustained eco-agriculture". Although the manuscript focuses on an important point, several concepts are badly explained. For instance:

- what is a sustained eco-agriculture? Sometimes, the authors write also restoration and bioremediation, and this subject is not clear enough. It seems the authors are trying to mix very different subjects without clearly explaining what is meant here.

- how can sRNAs contribute to such aims? This is not clear. Many of the names detailed in Table 1 are linked with enhanced tolerance. But to what? This is not clear. Finally, how the authors came up with these sRNAs (and not others) is also missing. Why wasn't a search made in GenBank or TAIR?

- the manuscript mixes sRNAs from plants and from microbial DNA. The logic beyond this is not well understood since many other sRNAs can help 

Finally, I'm not sure what is the novelty beyond this manuscript. I do not want to dismiss the authors work but there are already several reviews about the roles of sRNAs in enhancing tolerance. 

Author Response

Detailed responses to the Reviewers’ comments

Reviewer #3:

Thank you very much for your kind comments. We will try our best to revise the manuscript in order to improve its quality according to your suggestions and suit the requirements of the journal. Below are the itemized responses from authors to the comments of editor/reviewers (BLUE – Comments; Black – Authors response; Red – Revised text).

The manuscript aims to provide a review concerning the applications of small RNA for "a sustained eco-agriculture". Although the manuscript focuses on an important point, several concepts are badly explained. For instance:

The manuscript aims to provide a review concerning the applications of small RNA for "a sustained eco-agriculture". Although the manuscript focuses on an important point, several concepts are badly explained. For instance:

  1. what is a sustained eco-agriculture? Sometimes, the authors write also restoration and bioremediation, and this subject is not clear enough. It seems the authors are trying to mix very different subjects without clearly explaining what is meant here.

Thanks for your valuable comments and we are giving more importance to answer your comments. Accordingly, the sustained agriculture is farming in sustainable ways to meet the world demands without tampering the natural resources like soil quality, beneficial microbial community whereas the sustained eco-agriculture, serves all the above-mentioned purposes with an addition of protecting the eco system. The bioremediation, restoration and biomonitoring are the approaches are routes to achieve the sustained eco-agriculture where small RNA can be used as tools (Line 66-71).

  1. How can sRNAs contribute to such aims? This is not clear. Many of the names detailed in Table 1 are linked with enhanced tolerance. But to what? This is not clear. Finally, how the authors came up with these sRNAs (and not others) is also missing. Why wasn't a search made in GenBank or TAIR?

We apologize for the inconveniences as we have explained in our review, applications of sRNAs for abiotic and biotic stress tolerance externally is one of our hypothesized approach for the development of sustained eco-agriculture (Line 95-102) (Line 115-119). Yes, the Table 1 is showing the expression of tolerant sRNAs by plants to various stresses, which may be explored to develop sRNA-based fertilizers and pesticides. We surveyed the literature to obtain the sRNAs expressed by the plants in response to the abiotic and biotic stresses. Since it is a pilot study and first of its kind, we did not consider searching for the existing databases.

  1. The manuscript mixes sRNAs from plants and from microbial DNA. The logic beyond this is not well understood since many other sRNAs can help 

We agree with the reviewer we would like to bring it to your attention that this review intended to highlight the close interaction of plants and microbes for achieving a sustained eco-agricultural system, therefore we had discussed the specific small RNAs from plants and microbes (Line 104-108) (Line 111-114). We understand that the sRNAs from insects, parasites animals would also have an impact on the eco-agriculture, but there are limited data available and it would dilute the topic therefore we confined our analysis with only plants and microbes.

Round 2

Reviewer 3 Report

The authors have made several changes in the text to answer my comments/concerns. Thank you for your efforts in accomplishing this! I believe the manuscript reads very well.